# Characteristics and Functional Impact of Unplanned Acute Care Unit Readmissions during Inpatient Traumatic Brain Injury Rehabilitation: A Retrospective Cohort Study

**DOI:** 10.3390/life13081720

**Published:** 2023-08-10

**Authors:** Poo Lee Ong, Anna Rosiana, Karen Sui Geok Chua

**Affiliations:** 1Institute of Rehabilitation Excellence (IREx), Tan Tock Seng Hospital Rehabilitation Centre, Singapore 569766, Singapore; anna.rosiana@ttsh.com.sg (A.R.); karen_chua@ttsh.com.sg (K.S.G.C.); 2Lee Kong Chian School of Medicine, Nanyang Technological University, Singapore 639798, Singapore; 3Yong Loo Lin School of Medicine, National University of Singapore, Singapore 119077, Singapore

**Keywords:** rehabilitation, traumatic brain injury, acute care unit readmissions, head injury, functional independence measure, length of stay

## Abstract

Background: This study investigated the incidence, characteristics and functional outcomes associated with unplanned Acute Care Unit Readmissions (ACUR) during inpatient traumatic brain injury (TBI) rehabilitation in an Asian cohort. Methods: A retrospective review of electronic medical records from a single rehabilitation unit was conducted from 1 January 2012 to 31 December 2014. Inclusion criteria were first TBI, aged >18 years, admitted <6 months of TBI. ACUR were characterized into neurological, medical or neurosurgical subtypes. The main outcome measure was discharge and Functional Independence Measure (FIM™). Secondary outcomes included rehabilitation length of stay (RLOS). Results: Of 121 eligible TBI records, the incidence of ACUR was 14% (*n* = 17), comprising neurologic (76.5%) and medical (23.5%) subtypes occurring at median of 13 days (IQR 6, 28.5) after rehabilitation admission. Patients without ACUR had a significantly higher admission mean (SD) FIM score compared to those with ACUR (FIM ACUR-negative 63.4 (21.1) vs. FIM ACUR-positive 50.53(25.4), *p* = 0.026). Significantly lower discharge FIM was noted in those with ACUR compared to those without. (FIM ACUR-positive 65.8(31.4) vs. FIM ACUR-negative 85.4 (21.1), *p* = 0.023) Furthermore, a significant near-doubling of RLOS was noted in ACUR patients compared to non-ACUR counterparts (ACUR-positive median 55 days (IQR 34.50, 87.50) vs. ACUR-negative median 28 days (IQR 16.25, 40.00), *p* = 0.002). Conclusions: This study highlights the significant negative functional impact and lengthening of rehabilitation duration of ACUR on inpatient rehabilitation outcome for TBI.

## 1. Introduction

Traumatic brain injury (TBI) is a recognized global health problem of epidemic proportions with significant contributions to emergency department visits, hospitalizations and deaths. The estimated global incidence across all severities of TBI is approximately 939 cases per 100,000 people. Approximately 69 million individuals worldwide are estimated to sustain TBI each year, with varying degrees of functional impairments [1,2]. According to the CDC TBI surveillance report, there were more than 611 TBI-related hospitalizations and 190 TBI-related deaths per day [3].

Individuals who experienced moderate to severe TBI often need ongoing care and support as they progress in their recovery. The effects of TBI often extend beyond the individual’s life and severely impact their family and society, leading to health and economic burdens. The expenses associated with TBI encompass both direct and indirect medical costs; in 2010, these were estimated at approximately USD 76.5 billion [4].

Rehabilitation plays a crucial role in addressing the complex needs of individuals with moderate to severe TBI, facilitating functional recovery and improving long-term outcomes [5]. Specialized acute TBI inpatient rehabilitation programs aim to maximize independence, enhance quality of life and promote community reintegration for TBI survivors. 

TBI has also been associated with diverse medical complications for patients that are discharged from acute care hospital to inpatient rehabilitation settings. More than 80% of patients experienced at least one medical complication in cases of severe TBI. Hypertonia, agitation/aggression, urinary tract infection and sleep disturbances were among the most commonly reported problems during inpatient rehabilitation [6].

However, a subset of TBI patients experience frequent readmissions to acute care facilities whilst undergoing inpatient rehabilitation. These unplanned acute care unit readmissions (ACUR) are associated with adverse functional outcomes and increased healthcare costs, driven chiefly by the longer length of inpatient rehabilitation stay related to ACUR, as well as the higher institutionalization rates [7,8]. In addition, specialized interventions and treatments to manage ACURs also contribute to greater healthcare resource utilization. 

Previous studies have reported on the incidence, characteristics and impact of ACUR during TBI rehabilitation [7,8,9,10,11]. A multi-centre study in the United States and Canada [7] demonstrated that those with ACUR had longer rehabilitation lengths of stay and higher discharge rates to institutional settings. Similarly, a study from Italy [12] reported an almost doubled duration of rehabilitation length of stay, poorer functional outcomes and higher risk of mortality for those patients with ACUR. 

Regarding Asian TBI, there was limited literature regarding ACUR during TBI rehabilitation. Given the increasing incidence of fall-related brain injuries, particularly in India, China and other Asian countries, more research is needed on the Asian TBI population [13].

The primary objective of this study was to determine ACUR incidence and subtypes in an Asian tertiary rehabilitation centre and explore the relationships between ACUR, acute TBI characteristics, functional outcome and discharge placement. The discharge total Functional Independence Measure (FIM™) was the main primary outcome. We hypothesize that cultural, social and healthcare system factors may influence the patterns and consequences of ACUR during TBI rehabilitation. Through the identification of ACUR correlates and outcomes, patient stratification and improved care models could be implemented in areas ranging from acute hospital to inpatient rehabilitation settings.

## 2. Materials and Methods

### 2.1. Study Design

A retrospective review of the electronic medical records (EMRs) of patients admitted and discharged from a single inpatient TBI rehabilitation unit was performed from 1 January 2012 to 31 December 2014. Patients’ demographics, TBI characteristics, complications, hospital length of stay and functional data were independently extracted from a functional database registry recorded prospectively during inpatient rehabilitation. Institutional ethical approvals were obtained from the National Healthcare Group-NHG Domain Specific Review Boards (NHG DSRB 2018/01114) prior to data extraction. Informed consent was waived as the study involved used de-identified data and no human subjects were recruited.

### 2.2. Study Settings 

The study was conducted at Tan Tock Seng Hospital (TTSH) Rehabilitation Centre, Singapore, a tertiary centre, directly linked to the National Neuroscience Institute (NNI), Singapore, an acute neurosurgical unit and a level II trauma centre. Suitable TBI patients were admitted to TTSH rehabilitation centre for inpatient rehabilitation after screening by rehabilitation physicians. 

The TBI rehabilitation program at TTSH rehabilitation centre consists of rehabilitation therapies conducted for 3 h daily, delivered 5.5 days a week by an interdisciplinary team of physiotherapists, occupational therapists, speech therapists, nurses, social workers and psychologists. The programme consists of multiple components, including the management of disorders of consciousness, post-traumatic amnesia assessment and management, mobility and gait training, basic activities of daily living (ADL) training, cognitive assessment and retraining, rehabilitation nursing care, dietary and social work interventions, TBI psychoeducation and discharge planning. Functional Independence Measures™ (FIM™) [14] were recorded by FIM-certified rehabilitation therapists within 72 h of admission and discharge from rehabilitation.

### 2.3. Study Participants 

Inpatient EMRs were selected based on the following inclusion and exclusion criteria. Inclusion criteria were patients aged above 18 years with first-onset TBI confirmed by CT or MRI brain imaging and admitted within 6 months of TBI from acute neurosurgical services. Exclusion criteria were EMRs with non-TBI diagnoses (e.g., stroke, intracerebral haemorrhage, subarachnoid haemorrhage, arteriovenous malformation, tumours, infections), EMRs with missing admission or discharge FIM data. 

### 2.4. Data-Collection Procedures and Study Variables

Post-discharged inpatient EMRs were identified from Tan Tock Seng Hospital (TTSH) rehabilitation centre standing database registry (SDB # 2010/0039). Data were extracted without personal identifiers and used to construct a case record form consisting of two main data points, i.e., admission and discharge from inpatient TBI rehabilitation for both ACUR and non-ACUR patients.

Independent variables included for analysis were age, sex, race, employment status, presence of any pre-existing comorbidities, TBI mechanism, acute length of stay (LOS), duration and emergence from post-traumatic amnesia (PTA), which was measured using the Westmead PTA scale (WPTAS) [15], admission Glasgow Coma Scale (GCS), TBI management (surgical/conservative), ICU duration, complications from TBI (VP shunt, tracheostomy, skull fracture) and complications during inpatient rehabilitation. 

Dependent variables analyzed were days to ACUR after rehabilitation ward admission, reason for ACUR, FIM data, rehabilitation length of stay, discharge disposition, carer need and Glasgow Outcome Scale (GOS) on discharge. GOS was also classified as good outcome (4–5) and poor outcome (1–3). 

ACUR was defined as any occurrence of readmission to acute care facilities exceeding 24 h, primarily for medical or neurological reasons necessitating further treatment. ACUR due to elective reasons, such as surgical procedures or medical investigations, were not considered ACUR occurrence. 

Complications occurring during inpatient rehabilitation were defined as those that interrupted rehabilitation progress and/or required treatment. Not all these complications, including infection, cardiac issues, or neurological problems, necessitated ACUR although they warranted treatment. 

The duration of the acute care stay, referred to as Acute Length of Stay (Acute LOS), was defined as the time from admission to the acute care facility to admission to the rehabilitation centre. Rehabilitation Length of Stay (RLOS) was defined as the number of days between discharge and admission to rehabilitation centre. For patients with ACUR, their acute LOS during the period of ACUR was subtracted from their RLOS. 

### 2.5. Statistical Analysis

Statistical analyses were generated using Statistical Product and Service Solutions (SPSS) Version 26.0 (IBM Corp., Armonk, NY, USA). Descriptive statistics were utilized to illustrate patient demographics and clinical characteristics. Tests of normality were performed using Shapiro–Wilk test. Ordinal data were presented as means, SD, for normally distributed data, or medians, IQR, for non-normally distributed data. The distribution of categorical variables was compared using chi-square or Fisher’s exact test. A comparison of differences between groups for ordinal data was made using Mann–Whitney-U test. A *p* < 0.05 was considered statistically significant for a two-tailed test. 

## 3. Results

### 3.1. Baseline Demographic and TBI Characteristics 

Figure 1 illustrates the flow diagram for study recruitment.

A total of 131 medical records were screened and 121 records were included for analyses. 

Table 1 presents the demographics and characteristics of patients, comparing those with and without ACUR. 

The cohort’s mean age was 58.8 (SD 19.24) years and 72% (87) were males. The most common mechanism of injury at the time of admission was falls (64.5%), followed by road traffic accidents (27.3%). Among the 121 patients admitted for TBI, 17 (14%) experienced at least one ACUR, at a median of 13 days ± 14.19 after rehabilitation admission. In the study population, 69.4% had one comorbidity (*p* = 0.910) and there was no mortality recorded during rehabilitation.

Out of the study population, 84 patients (69.4%) had a mild GCS score ranging from 13 to 15 upon admission. However, it was observed that 78.5% of these patients still required caregiver support at the time of discharge. The majority of patients were discharged home from the acute care hospital (89.3%), while a smaller percentage (10.7%) were discharge to other health care facilities or acute hospitals.

### 3.2. ACUR Incidence, Classification and Acute Correlations

Table 2 summarizes the events leading to ACUR. 

The incidence of ACUR in our sample was 14% (17/121) and the most common reason was neurological (*n* = 13, 76.47%), whereas medical reasons accounted for 23.53% (*n* = 4).

There were no significant differences in baseline demographic, injury or acute management characteristics between ACUR and non-ACUR patients. The median acute hospital length of stay (ALOS) for all patients was 21 days, with a slightly higher median of 26 days for patients with ACUR (*p* = 0.247). Patients with ACUR had a significantly higher number VP shunts (*p* = 0.003) than those without ACUR (Table 1).

### 3.3. Relationships between ACUR and Functional Outcome

Table 3 presents a comparison of admission and discharge FIM outcomes by ACUR status. 

TBI patients without ACUR had significantly higher admission FIM, by 12.8 FIM points (mean 63.4 SD 21.1, vs. 50.5 (25.4), *p* = 0.026); FIM cognition was significantly higher (mean 19.1 SD 7.9, vs. 14.5 SD 8.2, *p* = 0.029), while FIM motor was similar between both groups (*p* = 0.059). 

Patients who experienced ACUR had a significantly poorer clinical outcome, as indicated by the lower discharge total FIM score, by 20 points (mean 65.8 SD 31.4, *p* = 0.023) compared to those without ACUR (mean 85.4 SD 21.1 *p* = 0.023). However, ACUR patients exhibited a significantly lower discharge score in the FIM motor domain (mean 47.2 SD 23.8) compared to those without ACUR (mean 63.1 SD 16.5, *p* = 0.016). In terms of the discharge FIM cognitive score, differences between ACUR-positive versus -negative patients were not significantly different, (ACUR-positive mean 18.5, SD 8.5 vs. ACUR-negative mean 22.2 SD 7.6, *p* = 0.069).

RLOS was, significantly, almost doubled in patients with ACUR (median 55 days (34.50–87.50)) compared to non-ACUR counterparts, (median 28 days (16.25–40.00), *p* = 0.002). The FIM gain for ACUR patients was 6 points lower than that for non-ACUR patients although this is not a statistically significant difference (ACUR mean 15.24 SD 23.59 vs. non-ACUR mean 21.99 SD 14.90, *p* = 0.117).

### 3.4. Factors Affecting Post-TBI Rehabilitation Outcome

Table 4 presents the various factors affecting post-TBI rehabilitation outcomes. TBI patients without ACUR demonstrated a significantly higher GOS score (GOS 4–5: 81 (78.6%) vs. 8 (50%), *p* = 0.026). The majority of TBI patients (*n* = 108, 89.3%) were discharged back to their homes, while 8.3% (*n* = 10) were discharged to nursing homes. There was no interaction between discharge placement and ACUR status. It is notable that, even among patients with mild GOS scores (GOS 4–5), a significant proportion (*n* = 95, 78.5%) still required care upon discharge, indicating ongoing support needs. Further subanalysis revealed that patients who were discharged home had a higher total discharge FIM score by 23 points compared to patients discharged to a nursing home (mean 85.5 SD 22.1 vs. mean 62.3 SD 23.9, *p* = 0.002). 

In terms of rehabilitation duration, ACUR patients experienced a significant increase in their length of stay compared to non-ACUR patients (55 days vs. 28 days, *p* = 0.002). 

On logistic regression analysis, an association was observed between the presence of VP shunt and the impact on ACUR, with an odds ratio of 0.079, *p* = 0.028, as well as admission FIM score, with an odds ratio of 0.970, *p* = 0.037. 

### 3.5. Correlations between Complications and Functional Outcome

Table 5 describes the functional outcomes in relation to the number of complications during inpatient TBI rehabilitation. Among the study cohort, 63 patients (52%) experienced at least one complication during their rehabilitation, with 21 patients (17.4%) encountering two or more complications. The most common medical complication observed during rehabilitation was urinary tract infection (UTI), accounting for 23.6% (*n* = 21/89) of all reported complications. 

Patients with at least one complication demonstrated significantly poorer functional outcomes. Their motor FIM score was lower by 8 points with one complication (*p* = 0.017) and 14 points with two complications (*p* = 0.017). Similarly, the total FIM on discharge was reduced by 6 points with one complication and 17 points with two complications (*p* = 0.049). 

For RLOS, this was extended by 6 days in patients with one complication and 39 days in those with two or more complications (*p* < 0.001).

## 4. Discussion

### 4.1. ACUR Incidence and Its Impacts on Functional Outcome and Length of Stay

Our study aimed to quantify the differences between ACUR and non-ACUR patients during inpatient TBI rehabilitation. We found that a total of 17 patients (14%) experienced ACUR in our population, which is consistent with the rates reported in other studies ranging from 9% to as high as 29.8% [7,12]. Such large variations could be related to differences in definitions and temporal diversity.

While previous reports suggested that ACUR occurrence was higher in older patients, surgically managed TBI, history of coronary artery disease, congestive heart failure, depression and the presence of dysphagia [7], our study did not find any significant correlations between demographic, injury or comorbidity and the occurrence of an ACUR. This indicates that other factors beyond the acute TBI or initial care phase may contribute to ACUR occurrence. Contributory factors were also that of a small cohort of relatively young and robust TBI patients (mean age of 58 years).

Primarily, neurological etiologies (76.5%) accounted for the majority of ACUR in our sample. This finding was consistent with 65% of the ACUR reported by Hammond et al. [7]. The higher ACUR incidence could be explained by systematic reasons, such as the rehabilitation centre having no on-site CT imaging or acute neurosurgical or ICU services; hence, ACUR was the preferred pathway of care to access expedient diagnostic imaging and consultation was needed to prevent further neurological deterioration. The relatively lower proportion of medically related ACUR could be attributed to the early intra-rehabilitation stay and medical management as long as patients were hemodynamically stable.

Our study revealed that patients with ACUR experienced poorer functional outcomes compared to those without ACUR, which was consistent with previous studies [7,12]. This was evident in the lower FIM admission scores and lower discharge FIM scores of the ACUR patients. However, there was no significant difference in FIM gains between the two groups, indicating comparable progress during rehabilitation despite initial differences in functional status. These results underscore the importance of functional abilities on admission, as patients with higher FIM motor scores at admission were less likely to experience ACUR [7]. These findings align with studies by Chung et al. [16] and Clinton et al. [13], both of which identified a lower admission FIM score as a significant predictor of higher ACUR rates.

We also observed a significant difference in the length of rehabilitation stay between patients with ACUR and those without, consistent with other studies [7,12]. ACUR patients had almost double the length of rehabilitation stay compared to non-ACUR patients, impacting direct healthcare costs. The reasons for this prolonged rehabilitation stay in the ACUR group could include the need for additional interventions, management of complications, truncation of rehabilitation during the ACUR period and secondary deconditioning.

Although statistically insignificant, the presence of ACUR in our study may be associated with a more severe initial injury, as indicated by the higher percentage of patients with severe GCS scores on admission in the ACUR group compared to the non-ACUR group (23.5% ACUR vs. 12.5% non-ACUR). This suggests that ACUR could potentially be a consequence of a more severe TBI, indirectly contributed by lower FIM cognition on admission. A more severe TBI is also associated with higher cytokine release and more multisystem complications [17]. This scenario may account for the observed poorer admission FIM score, lower FIM gain and longer rehabilitation stays in patients with ACUR.

### 4.2. Intra-Rehabilitation Complications and Its Associated Functional Outcomes

For the association between number of complications and functional outcomes, our study findings indicate that TBI patients without any complications and with only one complication experienced higher FIM gains with a median of 20 and 21 points compared to patients with two or more complications, with a median of 14.5 points (*p* = 0.121). However, all patients failed to meet the Minimal Clinically Important Differences (MCID) [18] of 25 for TBI (Table 4). These implies significant negative rehabilitation functional gains for patients facing multiple complications. Hence, it is imperative to consider modifying rehabilitation goals, implement targeted rehabilitation and extend rehabilitation LOS to achieve a comparable functional outcome once the first complications arise.

Common medical complications following the TBI studied previously in the acute care setting would include sepsis, respiratory infections, hypertension, acute kidney injury, diabetes, cardiac arrhythmias and extremity fractures [19,20,21,22]. Inpatient rehabilitation is also commonly associated with diverse disorders such as hydrocephalus, seizures, paroxysmal autonomic dysfunction, ventricular dilatation, abnormal liver function, hypertension, thrombophlebitis, respiratory infections, heterotopic ossification, fractures, pituitary-hypothalamic dysfunction, psychiatric and behavioural disturbances and issues with eyes, ears, nose and throat [23,24,25,26,27,28]. The high complication rates in acute TBI are proposed to be due to acute cytokine and chemokine release [17].

### 4.3. Discharge Placement and Caregiver Support

Despite the observed differences in functional outcome between ACUR patients, we found no significant difference in the need for caregiver support upon discharge or in discharge placement between the ACUR and non-ACUR groups, despite the lower discharge FIM in those with ACUR.

The majority of TBI patients in this study were discharged home irrespective of their ACUR status and functional outcomes. This finding contrasts with another study, where ACUR groups were associated with a higher likelihood of not being discharged home [7]. This may be due to the strong local community support and Asian social support, supporting our primary hypothesis.

### 4.4. Study Limitations

We highlight the following study limitations: a small sample size and retrospective study design from a single centre, which may restrict the generalizability of the findings to older or younger TBI populations and clinical preselection for admission to rehabilitation. Additionally, we were unable to ascertain the duration or nature of pre-existing comorbidities. Future research should aim to include diverse patient populations through multi-centre studies to improve the validity of the results.

## 5. Conclusions

In conclusion, our study provides evidence that ACUR during traumatic brain injury (TBI) rehabilitation is associated with poorer functional outcomes and a longer length of stay in rehabilitation. This highlights the importance of addressing and minimizing ACUR impacts to optimize patient outcomes and reduce socioeconomic burden. Reducing readmission to acute care facilities remains a significant challenge in TBI. This may be achieved by the early detection of complications, expedient management and ongoing rehabilitation therapy, although this should be modified during acute illness to reduce deconditioning and secondary decline. Future research should focus on identifying the underlying causes and risk factors associated with ACUR to develop targeted interventions and preventive strategies.

## Figures and Tables

**Figure 1 life-13-01720-f001:**
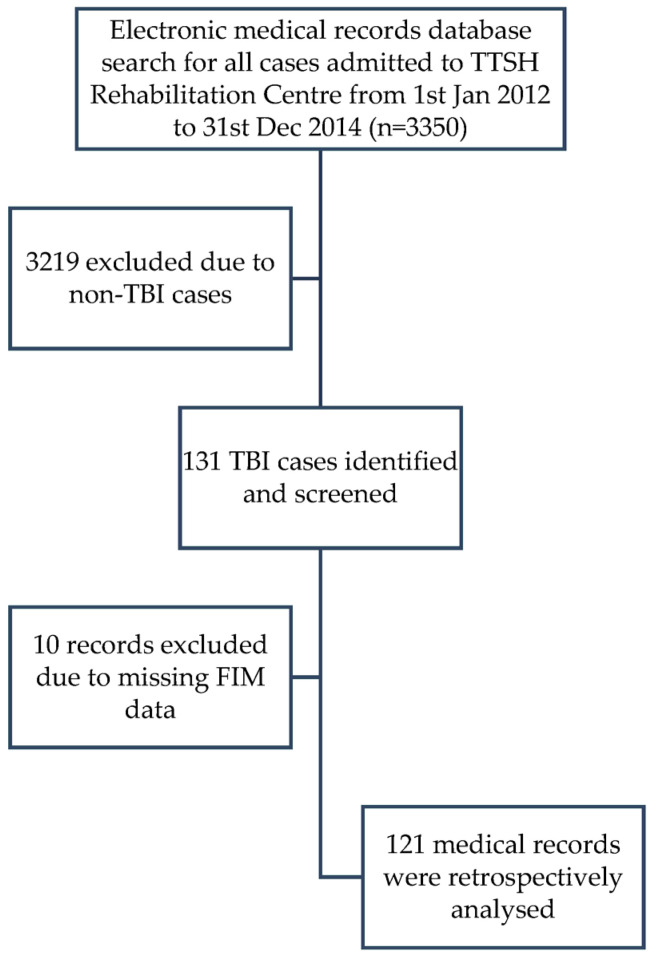
Flow diagram of study identification, inclusion and exclusion criteria.

**Table 1 life-13-01720-t001:** Demographics and characteristics between ACUR and non-ACUR group (*n* = 121).

Characteristic	Total (*n* = 121)	Non ACUR (*n* = 104)	ACUR (*n* = 17)	*p* Value
**Age**				
Age in years, mean (SD)	58.84 (19.24)	59.50 (19.25)	54.82 (19.29)	0.355 ^a^
**Sex, *n* (%)**				
Male	87 (71.9)	74 (71.2)	13 (76.5)	0.777 ^d^
Female	34 (28.1)	30 (28.8)	4 (23.5)	
**Race, *n* (%)**				
Chinese	103 (85.1)	89 (85.6)	14 (82.4)	0.462 ^d^
Malay	9 (7.4)	8 (7.7)	1 (5.9)	
Indian	6 (5.0)	4 (3.8)	2 (1.2)	
Others	3 (2.5)	3 (2.9)	0 (0)	
**Employment status *, (*n* = 83), *n* (%)**				
Employed	43 (51.8)	34 (47.9)	9 (75)	0.208 ^c^
Unemployed	40 (48.2)	37 (52.1)	3 (25)	
**TBI Mechanism, *n* (%)**				
Road traffic accident	33 (27.3)	28 (26.9)	5 (29.4)	0.385 ^d^
Fall	78 (64.5)	68 (65.4)	10 (58.8)	
Assault	6 (5.0)	4 (3.8)	2 (1.2)	
Others	4 (3.3)	4 (3.8)	0 (0.0)	
**Presence of comorbidity, *n* (%)**				
No	37 (30.6)	32 (30.8)	5 (29.4)	0.910 ^c^
Yes	84 (69.4)	72 (69.2)	12 (70.6)	
**Admission GCS, *n* (%)**				
3–8 (Severe)	17 (14.0)	13 (12.5)	4 (23.5)	0.361 ^d^
9–12 (Moderate)	20 (16.5)	17 (16.3)	3 (17.6)	
13–15 (Mild)	84 (69.4)	74 (71.2)	10 (58.8)	
**PTA Duration (days),**				
Median, (25th, 75th)	30.5 (18.8, 42.3)	30.0 (19.0, 42.0)	60.0 (36.0, 76.5)	0.324 ^b^
**Emergence from PTA upon discharge *, (*n* = 57), *n* (%)**				
Emerged	34 (59.6)	32 (60.3)	2 (50)	0.131 ^d^
Not emerged	23 (40.4)	21 (39.6)	2 (50)	
**TBI Management, *n* (%)**				
Surgical	57 (47.1)	49 (47.1)	8 (47.1)	0.950 ^c^
Conservative	64 (52.9)	55 (52.9)	9 (52.9)	
**Acute hospital LOS (days),**				
Median (25th, 75th)	21.0 (13.0, 32.5)	20.0 (12.3, 31.8)	26.0 (15.0, 45.5)	0.247 ^b^
**ICU duration, *n* (%)**				
>72 h	49 (42.2)	40 (40)	9 (56.3)	0.339 ^d^
<72 h	67 (59.3)	60 (60)	7 (43.8)	
**Presence of skull fracture, *n* (%)**				
Yes	63 (52.1)	53 (51.0)	10 (58.8)	0.608 ^c^
No	58 (47.9)	51 (49.0)	7 (41.2)	
**Presence of tracheostomy, *n* (%)**				
Yes	9 (7.4)	9 (8.6)	0 (0.0)	0.357 ^d^
No	112 (92.6)	95 (91.3)	17 (100)	
**Presence of VP shunt, *n* (%)**				
Yes	6 (5.0)	2 (2)	4 (23.5)	**0.003 ^d^**
No	115 (95.0)	102 (98)	13 (76.4)	

^a^ Independent samples *t* test, ^b^ Mann–Whitney U Test, ^c^ Pearson Chi Square Test, ^d^ Fisher Exact Test. missing data. Legends: TBI: traumatic brain injury, GCS: Glasgow coma scale, PTA: post-traumatic amnesia, LOS: length of stay, ICU: intensive care unit, VP shunt: ventricular peritoneal shunt. * indicates missing data for clarification.

**Table 2 life-13-01720-t002:** Classification of causes of ACUR episodes (*n* = 17).

Events Leading to ACUR	*n* (%)	
**Neurological**	13 (76.5)	
Hydrocephalus	3 (17.6)	
New intracerebral haemorrhage	3 (17.6)	
Seizure/epilepsy	2 (11.8)	
Worsening midline shift	2 (11.8)	
Stroke	1 (5.9)	
Cranial wound infection	1 (5.9)	
Sunken brain	1 (5.9)	
**Medical**	4 (23.5)	
Sepsis with unknown source	2 (11.8)	
Agitation and violent behaviour	1 (5.9)	
Autonomic dysfunction	1 (5.9)	
Total	17 (100%)	

**Table 3 life-13-01720-t003:** Comparison of functional outcome between ACUR and non-ACUR groups (*n* = 121).

Functional Outcome	Total (*n* = 121)	Non ACUR (*n* = 104)	ACUR (*n* = 17)	*p* Value
FIM (admission)				
Total, mean (SD)	61.6 (22.1)	63.4 (21.1)	50.5 (25.4)	**0.026 ^a^**
Motor, mean (SD)	43.1 (16.7)	44.3 (16.2)	36.0 (18.28)	0.059 ^a^
Cognition, mean (SD)	18.5 (8.04)	19.1 (7.9)	14.5 (8.2)	**0.029 ^a^**
FIM (discharge)				
Total, mean (SD)	82.6 (23.7)	85.4 (21.1)	65.8 (31.4)	**0.023 ^a^**
Motor, mean (SD)	60.9 (18.4)	63.1 (16.5)	47.2 (23.8)	0.016 ^a^
Cognition, mean (SD)	21.7 (7.8)	22.2 (7.6)	18.5 (8.5)	0.069 ^a^
FIM gain, mean (SD)	21.0 (16.4)	22.0 (14.9)	15.2 (23.6)	0.117 ^a^

^a^ Independent samples *t* test. Legend: FIM: Functional Independence Measure.

**Table 4 life-13-01720-t004:** Correlations between GOS, RLOS, carer need, discharge destination and ACUR status (*n* = 121).

**Variables**	**Total (*n* = 121)**	**Non ACUR** **(*n* = 104)**	**ACUR (*n* = 17)**	***p* Value**
GOS, *n* (%) *				
1–3	30 (24.8)	22 (21.4)	8 (50.0)	**0.026 ^a^**
4–5	89 (73.6)	81 (78.6)	8 (50.0)	
RLOS,	28.0 (17.0, 47.0)	28.0 (16.3, 40.0)	55.0 (34.5, 87.5)	**0.002 ^b^**
median (25th, 75th)				
Needed carer on discharge, *n* (%)				
Yes	95 (78.5)	82 (78.8)	13 (76.5)	0.759 ^a^
No	26 (21.5)	22 (21.2)	4 (23.5)	
Discharge destination, *n* (%)				
Own home	108 (89.3)	94 (90.3)	14 (82.4)	0.391 ^a^
Others	13 (10.7)	10 (9.6)	3 (17.6)	

^a^ Fisher Exact test. ^b^ Mann–Whitney U test. * Missing data. Legend: GOS: Glasgow Outcome Scale, RLOS: rehabilitation length of stay.

**Table 5 life-13-01720-t005:** Univariate analysis of complications and functional outcomes (*n* = 121).

Variables	No Complication (*n* = 58)	1 Complication (*n* = 42)	2 or More Complications (*n* = 21)	*p* Value
GOS, *n* (%)				
1–3	11 (36.7)	8 (26.7)	11 (36.7)	**0.004 ^c^**
4–5	47 (52.8)	33 (37.1)	9 (10.1)	
FIM (admission)				
Total, median (25th, 75th)	66.50 (50.00, 78.25)	61.50 (48.00, 75.50)	58.00 (20.00, 92.80)	0.327 ^b^
Motor, median (25th, 75th)	45.50 (31.75, 59.00)	43.00 (32.75, 51.00)	41.00 (15.00, 57.00)	0.251 ^b^
Cognition, median (25th, 75th)	20.00 (13.00, 23.25)	18.00 (14.50, 23.00)	13.00 (5.50, 24.50)	0.207 ^b^
FIM (discharge)				
Total, median (25th, 75th)	91.00 (74.50, 105.25)	85.00 (71.00, 99.25)	74.00 (40.50, 91.00)	0.049 ^b^
Motor, median (25th, 75th)	69.00 (53.00, 78.00)	61.00 (54.00, 72.25)	55.00 (28.00, 68.50)	**0.017** ^b^
Cognition, mean (SD)	21.78 (7.77)	22.86 (7.01)	19.24 (9.02)	0.220 ^a^
FIM gain				
Total, median (25th, 75th)	20 (12, 32)	21 (10, 33)	14.5 (4.5, 25.5)	0.121 ^b^
RLOS, median (25th, 75th)	24.50 (15.00, 38.00)	30.00 (17.25, 45.25)	63.00 (30.50, 99.00)	**<0.001 ^b^**

^a^ One way ANOVA, ^b^ Kruskal–Wallis test, ^c^ Pearson Chi Square test. Legend: GOS: Glasgow Outcome Scale, FIM: Functional Independence Measure.

## Data Availability

The data presented in this study are available on request from the corresponding author. The data are not publicly available due to institutional review board regulations.

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
