# Peer review of "Characteristics and Functional Impact of Unplanned Acute Care Unit Readmissions during Inpatient Traumatic Brain Injury Rehabilitation: A Retrospective Cohort Study"

_life, 2023, doi:10.3390/life13081720_

Round 1

Reviewer 1 Report

The paper is very interesting and good structured. There is a problem with the languange with many errors. But the contents and results are well presented

Extensive editing of English language required

Author Response

English language and manuscript edited and rechecked by colleague and all other study team members who are fluent in English

Reviewer 2 Report

1. A flow chart shows the enrollment, exclusion and exclusion in the study for total TBI patients who were admitted to TTSH rehabilitation center during the study period is necessary.

2. Table: There should be a definition of abbreviations in the legend.

3. Table 3: The FIM (discharge) data is in the wrong place.

4. The main limitation is too small number of patients. Only included 121 TBI patients and 17 acute care unit readmission. The study period lasted for 3 three years, maybe more patients could be included in the study.

Author Response

1. A flow chart shows the enrollment, exclusion and exclusion in the study for total TBI patients who were admitted to TTSH rehabilitation center during the study period is necessary.

Response: Flow chart added as Figure 1.

2. Table: There should be a definition of abbreviations in the legend.

Response: Legend added as table footer for all abbreviations.

3. Table 3: The FIM (discharge) data is in the wrong place.

Response: recheck and correct.

4. The main limitation is too small number of patients. Only included 121 TBI patients and 17 acute care unit readmission. The study period lasted for 3 three years, maybe more patients could be included in the study.

Response: Study team had screened through exhaustively on all cases admitted to rehabilitation centre over the study period. However only 121 TBI cases were eligible for analysis. Study team will consider to lengthen the study period to include more suitable cases for analysis in future.

Reviewer 3 Report

In this study, the author examined the occurrence, features, and functional outcomes related to unplanned readmissions to the Acute Care Unit (ACUR) during rehabilitation for traumatic brain injury (TBI) in an Asian group. Unplanned readmissions to the Acute Care Unit (ACUR) during traumatic brain injury (TBI) rehabilitation inpatients have a significant negative effect on functional outcomes and prolong the duration of rehabilitation. While data are overall presented, and writing are very logic, there are several minor points for the data, which needs author’s attention.

Minor scientific concerns:

1.     Line 182-182, some have space between p and value (p = 0.247), but some not (p=0.003). All should be consistent.

2.     Table 1-5, the labeling should be consistent. Some “n” is capital, but some not. In addition, some have space between value and parenthesis, but some not. All should be consistent.

Author Response

1. Line 182-182, some have space between p and value (p = 0.247), but some not (p=0.003). All should be consistent.

Response: recheck and corrected to ensure consistency.

2. Table 1-5, the labeling should be consistent. Some “n” is capital, but some not. In addition, some have space between value and parenthesis, but some not. All should be consistent.

Response: recheck on all table and relabel using small capital “n” for consistency.

Reviewer 4 Report

The manuscript is devoted to the important and interesting topic of TBI. The article is well written, the authors are careful and accurate in their conclusions and described in detail the limitations of the study, however, I would like to clarify some points:

1) It may be worth pointing out more clearly that ACUR may be a consequence of a more severe injury and a decrease in FIM and an increase in RLOS may not be a direct consequence of ACUR

2) IQR is not usually used as a descriptive statistic, since it is asymmetric relative to the median, it is preferable to use individual values of the 1st and 3rd quartiles

3) Was the multiplicity correction used and if not used, why?

4) It would be interesting to analyze the dynamics of FIM from admission to discharge using repeated measures GLM with the ACUR  presence as an intergroup factor

Author Response

1) It may be worth pointing out more clearly that ACUR may be a consequence of a more severe injury and a decrease in FIM and an increase in RLOS may not be a direct consequence of ACUR

Response: Added in discussion line 279-284

2) IQR is not usually used as a descriptive statistic, since it is asymmetric relative to the median, it is preferable to use individual values of the 1st and 3rd quartiles

Response: Replaced with 25th, 75th.

3) Was the multiplicity correction used and if not used, why?

Response: Not used.  Only independent variables were selected for regression analysis.

4) It would be interesting to analyze the dynamics of FIM from admission to discharge using repeated measures GLM with the ACUR presence as an intergroup factor

Response: study team had performed the analysis and found significant improvement for both group for FIM admission and discharge.

Round 2

Reviewer 2 Report

1. Table 1: The data on employment status and PTA duration were in the wrong place.

2. Table 1: TBI management P value > 0.950c. The symbol “>” is not necessary.

Author Response

1. Table 1: The data on employment status and PTA duration were in the wrong place.

Response : Edited data on table 1 for both employment status and PTA duration to correct row. 

2. Table 1: TBI management P value > 0.950c.The symbol “>” is not necessary.

Response: ">" removed at p value of TBI management.